# Enriched Environment Attenuates Enhanced Trait Anxiety in Association with Normalization of Aberrant Neuro-Inflammatory Events

**DOI:** 10.3390/ijms232113052

**Published:** 2022-10-27

**Authors:** Anupam Sah, Sinead Rooney, Maria Kharitonova, Simone B. Sartori, Susanne A. Wolf, Nicolas Singewald

**Affiliations:** 1Department of Pharmacology and Toxicology, Institute of Pharmacy and Center for Molecular Biosciences Innsbruck (CMBI), University of Innsbruck, Innrain 80-82/III, A-6020 Innsbruck, Austria; 2Cellular Neurocience, Max-Delbrueck-Center for Molecular Medicine in the Helmholtz Association, 13125 Berlin, Germany; 3Department of Experimental Ophthalmology, Charité-Universitätsmedizin Berlin, 13353 Berlin, Germany

**Keywords:** environmental enrichment, trait anxiety, innate anxiety, stress, anxiolytics, inflammation, microglia, dentate gyrus, hippocampus, medial prefrontal cortex

## Abstract

Neuroinflammation is discussed to play a role in specific subgroups of different psychiatric disorders, including anxiety disorders. We have previously shown that a mouse model of trait anxiety (HAB) displays enhanced microglial density and phagocytic activity in key regions of anxiety circuits compared to normal-anxiety controls (NAB). Using minocycline, we provided causal evidence that reducing microglial activation within the dentate gyrus (DG) attenuated enhanced anxiety in HABs. Besides pharmacological intervention, “positive environmental stimuli”, which have the advantage of exerting no side-effects, have been shown to modulate inflammation-related markers in human beings. Therefore, we now investigated whether environmental enrichment (EE) would be sufficient to modulate upregulated neuroinflammation in high-anxiety HABs. We show for the first time that EE can indeed attenuate enhanced trait anxiety, even when presented as late as adulthood. We further found that EE-induced anxiolysis was associated with the attenuation of enhanced microglial density (using Iba-1 as the marker) in the DG and medial prefrontal cortex. Additionally, EE reduced Iba1 + CD68+ microglia density within the anterior DG. Hence, the successful attenuation of trait anxiety by EE was associated in part with the normalization of neuro-inflammatory imbalances. These results suggest that pharmacological and/or positive behavioral therapies triggering microglia-targeted anti-inflammatory effects could be promising as novel alternatives or complimentary anxiolytic therapeutic approaches in specific subgroups of individuals predisposed to trait anxiety.

## 1. Introduction

Anxiety disorders are the most prevalent mental illnesses. According to the World Health Organization, 3.6 percent—or about 264 million individuals worldwide—have an anxiety disorder. (https://apps.who.int/iris/bitstream/handle/10665/254610/WHO-MSD-MER-2017.2-eng.pdf?sequence=1 (accessed on 20 September 2022)). Additionally, a recent study has shown that the COVID-19 pandemic further led to a 25.6% increase in cases of anxiety disorders globally [1]. Genetic predisposition to high-trait anxiety has been identified as a severe risk factor for anxiety disorders and/or depression in later life [2]. New treatment approaches for such pathologies are necessary to overcome problems such as treatment resistance, high suicide risk, and treatment complications by comorbidities (for review, see [3]). However, this requires a better understanding of underlying pathophysiological mechanisms [4]. Neuroinflammation has recently been recognized as a potential mechanism contributing to the onset and/or maintenance of psychiatric disorders including depression and anxiety, as well as resulting in resistance to current treatments [5,6]. Neuroinflammation in the CNS involves key factors, such as microglial migration and activation, and can exert detrimental or beneficial consequences within an organism [7]. Specifically, the presence of high inflammation and associated dysregulated downstream pathways has been linked with treatment resistance to antidepressants [8], which are currently used as first-line treatment in many anxiety disorders. Immune-targeting interventions have thus been proposed as an alternative or add-on route in the treatment of psychiatric disorders [9].

Environmental conditions (such as noise, air pollution, weather conditions, housing conditions) form important components influencing mental health, including anxiety disorders. Environmental exposures can trigger mental disorders or represent protective factors by facilitating stress reduction, mental recovery, etc. [10,11,12]. Indeed, a meta-analysis study showed that exposure to natural environments was associated with a moderate increase in positive affect and a smaller, yet consistent, decrease in negative affect relative to comparison conditions [13,14]. Additionally, lower immune activation was observed following social stress in rural compared to urban participants raised with regular or no animal contact, respectively [15]. A functional magnetic resonance imaging study in human beings demonstrated that environmental conditions differentially activated brain regions, some of which are important nodes in the anxiety circuit. For example, viewing rural living environment photos activated the anterior cingulate gyrus and globus pallidus, whereas urban photo viewing activated the hippocampal and amygdalar regions (blood-oxygen-level dependent signal, [16]. Together these findings suggest that environments of varying stress, complexity and novelty exert a profound effect on structure and activity of anxiety-related brain regions, and thus can be either beneficial or detrimental in outcome.

Enriched environments (EEs) can be used as a therapeutic and have been shown to exert beneficial effects both in human beings and rodents [17,18]. EEs have also been shown to reduce reactivity to stress and anxiety (for review, see [19], increase cognitive function [20], and enhance learning and memory mechanisms [21,22]. Environmental therapeutic strategies, such as EEs, have an advantage over pharmacological approaches as they exert virtually negligible adverse (side) effects. Specifically in relation to anxiety, EE housing in rodents has been shown to reduce stress-induced anxiety (e.g., [23]).

The current study, therefore, used a mouse model which exhibits high-trait anxiety (HAB) compared to normal-anxiety (NAB) controls. This phenotype is the result of a selective breeding approach of CD-1 mice, according to anxiety scores assessed in the elevated plus maze test [24]. We tested whether and how beneficial (EE) environmental manipulations are capable of attenuating hyperanxiety. We observed that even inborn and seemingly rigid behavioral and neuroendocrine phenotypes can be rescued by positive environmental stimuli presented at an early developmental period [25]. It still remains to be investigated whether EE can mitigate genetically determined hyperanxiety when presented during adulthood.

Multiple mechanisms of EE-induced anxiolysis has been studied in chronic stress models, and there have been numerous studies demonstrating that EEs enhance neurogenesis [26,27,28] and induces an anti-inflammatory state or inhibition of microglia activation (for review, see [29,30]). On the other hand, only few studies have assessed the role of inflammation in the absence of stress so far. We have recently shown that high-anxiety HAB mice display clear signs of a central immune dysregulation, indicated by an increased microglial density and coverage in a specific set of brain regions, including the dentate gyrus, cingulate cortex, basolateral amygdala, paraventricular nucleus of the hypothalamus and the nucleus accumbens, brain regions that are part of anxiety and/or depression circuits [31]. Furthermore, we were able to show that chronic orally administered minocycline attenuated HAB hyperanxiety, which was associated with reduction in microglial density as well as phagocytic activity within the dentate gyrus [31]. Therefore, in the current study we aimed to test whether a non-pharmacological approach could sometimes mimic the minocycline treatment and investigated whether EE-induced anxiolysis is associated with normalization of aberrant central immune dysregulation in HABs.

## 2. Results

### 2.1. EE Housing Attenuates Anxiety Behavior in HAB

The HAB mice were chronically housed in the EE and subsequently tested for anxiety-like behavior, compared to the HAB counterparts housed in standard cages. OF was conducted following 4 weeks of the EE. In the OF test (Day 27), the HAB-EE group displayed an increased amount of time spent in the center of the arena (t(22) = 3.692, *p* < 0.01; Figure 1a) and number of entries into the center arena (t(22) = 4.315, *p* < 0.001; Figure 1b) compared to the standard-housed HABs. There was a slight trend toward increased total distance traveled in the HAB-EE group (t(23) = 1.850, *p* = 0.08; Figure 1c). The Mann–Whitney test revealed that the HAB-EE group displayed a significantly earlier latency for first entry to the center compared to the HAB-Std (U = 25.00, *p* < 0.01; Figure 1d). These data indicate that EEs induce an anxiolytic-like effect in HABs. 

To support the findings of the OF test, we subjected the mice to another anxiety-test, namely the LD test on the next day. The HAB mice housed in the EE displayed an increased amount of time spent in (t(22) = 6.195, *p* < 0.001; Figure 1e), number of entries to (t(22) = 4.768, *p* < 0.001; Figure 1f), and distance traveled in (t(22) = 5.273, *p* < 0.001; Figure 1g) the light arena compared to the standard-housed HABs. The Mann–Whitney test revealed that the HAB-EE group displayed lower latency for the first entry into the light zone compared to the HAB-Std (U = 22.00, *p* < 0.01; Figure 1h). Therefore, the findings obtained in the LD test confirmed the findings in the OF test.

### 2.2. EE-Induced Anxiolysis Is Associated with Attenuation of Aberrant Microglia in HABs within the Anterior and Posterior DG

We have previously shown that the HAB mice display enhanced microglial expression in the DG and mPFC, compared to the NAB controls [31]. In the current study, we observed that chronic EE housing reduced Iba1+ microglia density (t(19) = 3.401, *p* < 0.01; Figure 2a) and the percentage of coverage within the anterior DG (t(20) = 2.234, *p* < 0.05; Figure 2b) compared to the HAB-Std. Additionally, the microglial density (r = −0.49, *p* < 0.05; Figure 2c) was negatively correlated with time spent in the light compartment of the LD test.

Likewise, Iba1+ microglia density (t(22) = 2.923, *p* < 0.01; Figure 3a) and the percentage of coverage of the posterior DG (t(22) = 4.565, *p* < 0.001; Figure 3b) were significantly attenuated in the posterior DG of HAB-EE vs. HAB-Std mice.

### 2.3. EE-Induced Anxiolysis Is Associated with Attenuation of Phagocytic Microglial Density in HABs within the DG

CD68 is a marker for phagocytosis/antigen-presentation. We have previously shown that there is an increased density of co-labeled Iba1 + CD68+ cells in the granular cell layer of HAB compared to NAB, indicating a potential increase in microglial phagocytic activation in this region in HAB compared to NAB mice [31]. Furthermore, we also found that chronic oral minocycline indeed reduced HAB hyperanxiety, which was associated with significant decreases in Iba1+ and CD68 + Iba1+ cell densities in the DG. Paralleling the findings with minocycline, we now observe that EE-induced anxiolysis was associated with an attenuation of co-labeled Iba1 + CD68+ microglia density in the anterior DG (t(20) = 3.017, *p* < 0.01; Figure 4a) and in the posterior DG (t(22) = 2.737, *p* < 0.05; Figure 4b) in EE- vs. standard-housed HAB mice.

### 2.4. EE-Induced Anxiolysis Is Associated with Attenuation of Upregulated Microglia in HABs within the mPFC

Apart from the DG, we also observed in our previous study that microglia density was higher in the mPFC of HAB as compared to the NAB controls [31]. Therefore, we also investigated whether the reduced anxiety in adult HAB mice following the EE is mediated in part via attenuating aberrant neuro-inflammatory events in the mPFC. Indeed, within the mPFC, chronic EE housing reduced the previously demonstrated enhancement of Iba1+ microglia density (t(22) = 2.916, *p* < 0.01; Figure 5a). There was a trend toward a reduction in percentage of Iba1 coverage (t(21) = 1.753, *p* = 0.094; Figure 5b) in the HAB-EE group compared to the standard-housed HAB controls.

## 3. Discussion

We have previously shown that enhanced trait anxiety is associated with aberrant neuro-inflammatory events in key regions of the anxiety circuit within the CNS. We now investigated whether a non-pharmacological intervention, namely EEs previously shown to normalize hyperanxiety, utilizes neuro-inflammatory mechanisms in exerting their beneficial anxiolytic effect. We are able to show for the first time that indeed beneficial environmental changes have the capacity to dampen inflammatory events in association with alleviating trait hyperanxiety in genetically predisposed individuals. 

### 3.1. EE-Mediated Anxiolysis in HAB Mice

Extending the finding that a positive housing environment (EE) during early life is able to shift the anxiety behavior of male HABs [25,32], we now show that EE-onset at adulthood is capable of attenuating trait anxiety even during adulthood, reflecting the robustness of positive intervention in mitigating trait anxiety. 

We observed that HABs exposed to the EE displayed a slightly enhanced overall locomotion (Figure 1c) in the open-field test. One might argue that the enhanced locomotion could confound the processing of anxiety parameters in the open field and the light–dark tests. First, the effect is very small and second, we have previously shown that HAB-EE display similar levels of activity in the familiar home cage environment when compared against the standard housing HABs [25]. Thus, the enhanced locomotion observed in the unfamiliar open field might reflect an enhanced novelty seeking behavior induced by the anxiolytic properties of the EE. 

Despite the heterogeneous nature of the EE and its mechanisms, one consistent pattern that emerged from animal models of stress-induced anxiety in the past and now from animal models of trait anxiety is that the EE affects parts of the stress anxiety circuits such as the mPFC [33] or DG [34] and fine-tune the ability to respond appropriately to various challenges [35,36], (for review see [19]). This observation is in line with the conceptual framework of the inoculation stress hypothesis, which proposes that repeated exposure to novel, diverse stimuli in the EE prepares an individual to cope with future stress [37].

The use of EEs in the current study is important from a translational perspective, since investigating the role of the positive affect system may be utilized as an important yet underexplored treatment target in anxiety and depression. Existing interventions primarily target the negative affect system, yielding only modest effects on measures of positive emotions and associated outcomes (e.g., psychological well-being) [38]. 

### 3.2. EE Attenuates Aberrant Microglial Phenotype in the HAB Mice

We and others have previously shown that both male HAB mice and rats display on aberrant microglial phenotype in both the anterior and posterior DG as well as the mPFC [31,39]. Therefore, we next wanted to investigate whether an EE-induced anxiolytic-like effect in adult HAB mice alleviates the aforementioned aberrant neuroinflammation in the mPFC and DG. In various animal models, the EE has been shown to alter a range of inflammatory mechanisms [30], exerting an anti-inflammatory or microglial “activation” inhibition [40,41,42], but not yet been studied in a trait anxiety model. We now show that the EE reduces the enhanced microglia density in the mPFC and DG of HAB mice, in association with anxiolysis. Interestingly in “normal anxiety” mice sometimes the opposite effect, or lack thereof, has been shown in the literature. For example, the EE even increases the microglia number in the DG or hippocampus [43,44,45], or otherwise has shown no effect [46,47]. However, adverse or stressful environments that increase anxiety behavior are mostly reported to increase the hippocampal microglia number and/or activation (e.g.,: isolation housing, [47]; chronic mild stress, [48,49]; high-fat diet, [50,51]. Along with our findings, whether EEs elicit effects on microglia depend on the pre-existing pathology. Indeed, in support of this idea, a study by Piazza and colleagues demonstrated that diabetic rats displayed enhanced microglia in the DG compared to the non-diabetic controls, and exposure to EE housing decreased microglial activation in the DG [43]. It is of note that the diabetic mouse models also show comorbid anxiety and neuroinflammation (e.g., [52]). Finally, another study also showed that long-term EEs reduce microglial morphological diversity within the DG [53]. Besides the DG, another study also showed that microglia within the mPFC is altered in response to microbial intervention that is associated with changes in anxiety-like behavior [54]. Taken together, our results support the hypothesis that EEs reduce anxiety in a high-trait anxiety model at least partly via attenuation of the enhanced microglial expression in the key anxiety-brain regions. It is important to note that EEs attenuate enhanced anxiety as well as upregulated microglial expression in HABs. However, whether EEs are sufficient to normalize the enhanced anxiety as well as dysregulated microglia is still not known and will be addressed in future studies. Moreover, while we have observed EE-induced attenuation in microglial density as well as coverage, it remains to be investigated whether the microglial morphology both at baseline and following EEs is altered. This further characterization will be carried out using Hierarchial cluster analysis in future study. Thus, together with our pharmacology data reported earlier, the findings of the current study suggest that beneficial environment targeting the microglial system can alleviate trait anxiety symptoms in individuals via anti-inflammatory mechanisms in the CNS.

Microglia are regulators of neurogenic activity [55] and suppressed neurogenesis following acute stress has been associated with an increased presence of microglia in the DG [56]. In a recent study, it has been shown that microglia are critical for EE-mediated augmentation of neurogenesis: microglia depletion, via PLX5622, increased EE-dependent neurogenic activity in the DG, in association with reduced anxiety in a transgenic mouse model of Alzheimer’s disease [57]. Thus, taken together with our previous reports that neurogenesis as well as neuronal activation is suppressed in the DG of HABs [58], it may be suggested that there is a link between reduced hippocampal neurogenesis and enhanced microglial density/ phagocytosis in high-trait anxiety. An increase in density and/or “activation” state of hippocampal microglia is frequently associated with suppression of neurogenesis also in various stress models, for example acute unpredictable stress [59], inflammatory pain [60], transgenic model for neuroinflammation and allergy [61,62]. Thus, the evidence presented here suggests that microglia-mediated effects on neurogenesis in the DG may play a (partial) role in the maintenance of hyperanxiety in adult HAB mice. 

### 3.3. Effects of EE on Microglial CD68-Mediated Phagocytosis in HAB Mice

Specific cellular markers expressed on microglia are important indicators of microglial activity in the central nervous system (CNS). Using double labeling of Iba-+ and CD68+ [63], we have previously shown the increased presence of phagocytic microglia cells in the DG of HAB [31]. Therefore, besides investigating Iba-1 as a marker for microglia [64], we next aimed to determine the expression levels of the phagocytosis/antigen-presentation marker CD68 within Iba-1+ cells in the DG. We now demonstrate that the EE reduces CD68 expression in microglia in the DG of HAB in association with successful anxiolysis. In support of a link between hippocampal CD68 and hyperanxiety, Abuelezz and colleagues found that adverse environmental stress (chronic unpredictable mild stress, CUMS) increases CD68+ cell counts in the hippocampus in association with increased anxiety behavior. Consequently, upon systemic administration of the anti-oxidant and anti-inflammatory, coenzyme-Q10 displayed a reduction in CUMS-induced anxiety and in hippocampal CD68+ cell count [65]. Specific to the use of EEs, a recent study has shown that EEs prevent the age-related increase in CD68 expression in the hippocampus, in association with improved spatial memory in the Morris water maze [66]. Along these lines, our results suggest that alleviating anxiety by EEs may be (or at least partially) due to an EE-mediated suppression of microglia phagocytic activity in the DG, potentially also influencing neurogenesis (see above). In support of this idea, EEs decreased Iba1 expression in the DG in association with increasing BrdU+ cell count in the same brain region of diabetic rats [43]. Based on this indirect evidence, future studies are planned to investigate a direct link between EE-mediated anxiolysis in HAB and neurogenesis-microglia mechanisms.

## 4. Materials and Methods

### 4.1. Animals

Adult male HAB mice (*n* = 12–13 per group) were selectively inbred for their specific anxiety-related behavior at the Department of Pharmacology, Innsbruck Medical University, Innsbruck (Austria). The phenotype of each subject was confirmed by scores on the elevated plus maze (EPM) at 7 w of age, prior to any experiments [58]. HABs and NABs had access to food pellets (SSNIFF standard mouse chow) and water ad libitum and were group-housed in individually ventilated cages under standard laboratory conditions (12:12 light/dark cycle with lights on at 07:00 h, 22 ± 2 °C, 45–60% humidity). All experiments were approved by the Austrian Animal Experimentation Ethics Board (Bundesministerium für Wissenschaft Forschung und Wirtschaft, Kommission für Tierversuchsangelegenheiten) and were in compliance with international laws and policies. 

### 4.2. Enriched Environment

HAB mice were housed in enriched environment (EE) cages for 28d (3–5 animals per cage) (Figure 6b) from 12 w of age. In contrast to standard (Std) caging (Figure 6a) (Tecniplast type II, dimensions 331 × 159 × 132 mm, floor area 335 cm^2^), EE cages consisted of a bigger cage (Tecniplast type IV, dimensions 480 × 375 × 210 mm, floor area 1500 cm^2^), extra bedding material, and various toys including a transparent tunnel (Ferplast; 6 × 11 cm), transparent T-shaped tunnel (Ferplast; 6 × 14 × 10 cm), rope ladder (Trixie; 9 cm length), wood bridge (Trixie; 22 × 10 cm), wood suspension bridge (Trixie; 7 × 55 cm) and ping pong ball (4 cm diameter). Running wheel was excluded as to avoid monopolization of exercise opportunities. Bedding was changed twice a week. After 28d of the EE, mice were subjected to open field (OF) and light–dark (LD) test, and perfused and sacrificed 2 h post-LD test [24].

Following 4 weeks of the EE, mice were subjected to open field (OF) and light–dark (LD) tests according to previous protocols [24]. During the tests the animals were recorded with a video camera centrally positioned above the testing arena. The behaviors displayed were automatically analyzed with Ethovision (Ethovision 11 XT, Noldus, Wageningen, The Netherlands).

### 4.3. Open-Field Test

The OF test took place on day 27 of the EE. The test arena was an open box (50 cm × 50 cm) and the central compartment of the OF was illuminated at 150 lux. Mice were individually placed into the periphery of the OF with the nose facing one of the corners and allowed to explore the full arena for 10 min. The following anxiety-related parameters were analyzed: total distance traveled, time spent in the center, entries to the center and latency to first entry to center. 

### 4.4. Light–Dark Test

The light–dark test was performed on day 28 of the EE. The test arena consisted of an open box (50 cm × 50 cm), and a black box with a closed roof that was inserted covering 1/3 of the arena. The light arena was illuminated at 300 lux, and a closed dark chamber at <10 lux that was accessible through a small door (7 cm × 7 cm wide) assigned as a transition zone. The testing was conducted in accordance with established laboratory experimental protocols [67]. In brief, mice were individually placed with the nose facing the door of the dark chamber and, when it entered the dark chamber, it was allowed to explore the full test arena for 10 min. Parameters measured included time spent in, number of entries to, distance traveled in, and latency of first entry to the light arena.

### 4.5. Immunohistochemistry

At 2 h following the light–dark test, all mice were terminally anesthetized with an overdose of thiopental (200 mg/kg i.p.) and the chest cavity opened. Animals were transcardially perfused with 0.9% saline followed by 4% paraformaldehyde in phosphate buffer solution (pH 7.4), as previously described [58]. Brains were dissected out and temporarily post-fixed in 4% paraformaldehyde solution for 2 h, followed by permanent storage in phosphate buffer (0.1 mol/L pH 7.4). Coronal sections (40 μm) were cut using a vibratome (Leica VT 1000S, Nussloch, Germany). Fluorescent immunohistochemistry was performed on free-floating sections, which were blocked for 1 h in FSGB + T (1% BSA, 0.2% fish skin gelatin and 0.1% Triton X-100 in phosphate-buffered saline (PBS), all from Sigma, Vienna, Austria) and incubated overnight at room temperature (RT) with goat anti-Iba1 (1:500 Abcam, Abcam, Cambridge, United Kingdom, Cat# ab 107159) and rabbit anti-CD68 (1:300 Abcam, Cat# ab 125212) primary antibodies diluted in PBS, using a standard immunohistochemistry protocol [58]. This was followed by washing in PBS and incubation with secondary donkey anti-goat Alexa Fluor^®^ 647 (1:1000 Jackson Immuno Research, Vienna, Austria, Cat# ab 2340436) and donkey anti-rabbit Alexa Fluor^®^ 568 (1:1000, Invitrogen, Vienna, Austria, Cat# A10042) antibodies for 3 h at RT. The sections were then washed in PBS, mounted onto microscope glass slides (76 × 26 mm, frosted; Carl Roth, Germany, Cat# H868) and air-dried. Finally, slides were cover-slipped with ProlongTM Gold Antifade Mountant with 4′,6-diamidino-2-phenylindole (DAPI; Invitrogen, Cat# P36930).

### 4.6. Immunofluorescence Microscopy

We agree with the reviewer and have therefore further elaborated on the methodology section: Three consecutive sections each from the anterior and posterior consisting of polymorphic, hilus and granular cell layers, identified with Franklin & Paxinos Mouse Brain Atlas (3rd Ed.; [68]), were taken using a fluorescent microscope (Olympus BX51 microscope, Vienna, Austria). A 4× and subsequently 10× objective was applied to locate specific brain structures and the images were taken at 20× for quantitative analysis. Additionally, images of two consecutive sections were taken from the medial prefrontal cortex (mPFC). Quantification of immunopositive-cells was performed in both the left and right hemispheres in each of the brain sections using Image J software (National Institue of Health (NIH), Bethesda, MD, USA). Specifically, Image J was used to objectively quantify Iba1+ cell density (count/mm^2^) and % Iba1 coverage of the representative area. Coverage is defined as the sum total of the area of all cells in a given region and expressed as a fraction of the total area of the given region in % age. An Iba1+ cell was considered CD68+ when Iba1 was spatially co-expressed with a positive CD68 stain and were manually quantified in representative areas.

### 4.7. Statistical Analysis

Data analyses were performed with GraphPad Prism 5.0 software (GraphPad Software Inc., San Diego, CA, USA), following exclusion of outliers identified by Grubb’s test. D’Agostino-Pearson normality test was used to assess normal distribution of the data sets. Data were analyzed by unpaired Student’s *t*-test (two-tailed) for two group comparisons. Non-parametric data were analyzed by the Mann–Whitney test. Significance was set at *p* < 0.05, and data are presented as mean + standard error of the mean (S.E.M.).

## 5. Conclusions

We show here for the first time that EEs can ameliorate enhanced trait anxiety during adulthood, suggesting that positive behavioral therapies are robust enough to counter manifested hyper-anxiety. We further demonstrate that this anxiolysis occurred in association with the modulation of an inflammatory dysregulation in HAB; EEs significantly attenuated the enhanced microglia density and phagocytic activity in key regions of the anxiety circuit in the HAB mice. Thus, the present data extend our previous findings and suggest that pharmacological as well as positive behavioral therapies exerting microglia-targeted anti-inflammatory properties could be promising as novel alternatives or complimentary anxiolytic therapeutic approaches in specific subgroups of individuals predisposed to trait anxiety.

## Figures and Tables

**Figure 1 ijms-23-13052-f001:**
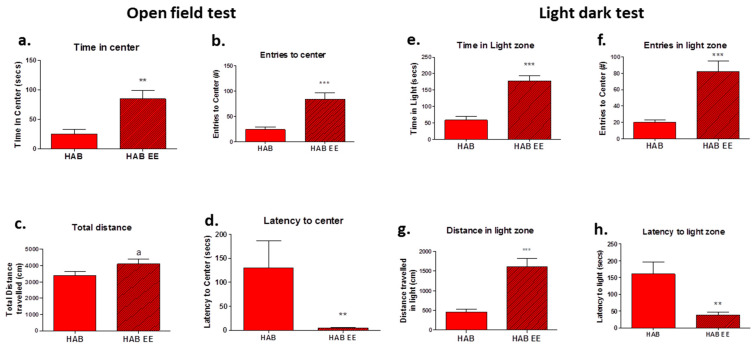
Chronic EE housing attenuates anxiety in HAB mice. In the OF test, HAB-EE group spent significantly more time in, made more entries to, and exhibited less latency to first entry to, the center (**a**,**b**,**d**), compared to standard-housed HAB. There was a trend toward greater distance traveled in HAB-EE (**c**). In the LD test, HAB-EE group spent significantly more time in the light, made more entries to the light, traveled more distance in the light, and exhibited earlier latency to first enter the light (**e**–**h**), compared to standard-housed HAB. Data are presented as mean ± SEM. *n* = 12–13. ^a^
*p* = 0.08, ** *p* < 0.01, *** *p* < 0.001 (Student *t*-test).

**Figure 2 ijms-23-13052-f002:**
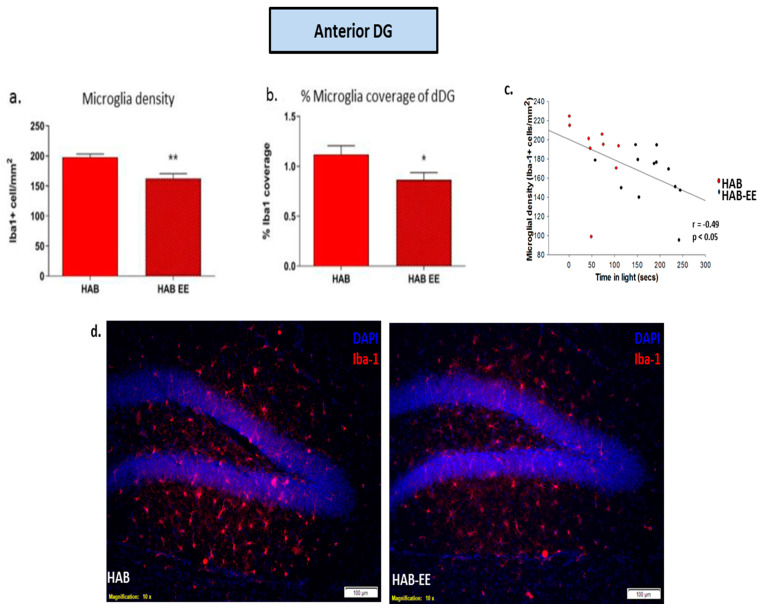
Iba1-related parameters in the anterior dentate gyrus are altered in HAB by EE housing. Iba1+ cell density and percentage of coverage were attenuated in HABs housed in enriched environment conditions (HAB-EE) compared to HABs housed in standard conditions (**a**,**b**). An increase in time spent in the light zone of the light–dark test was correlated with a reduction in microglial density (**c**). Representative images of Iba1 in the anterior DG, scale bar 100 µm (**d**). Data are presented as mean ± SEM. *n* = 12–13. * *p* < 0.05, ** *p* < 0.01.

**Figure 3 ijms-23-13052-f003:**
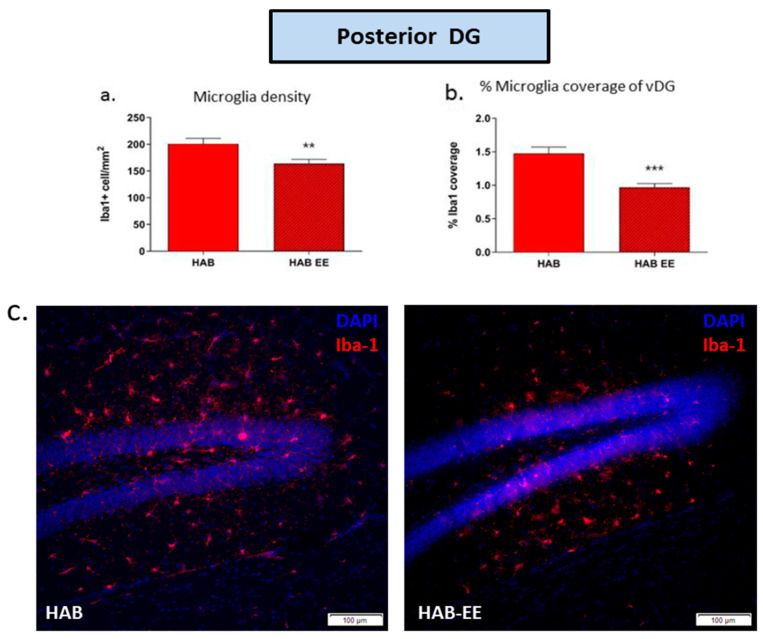
Iba1-related parameters in the posterior dentate gyrus were altered in HAB by EE housing. Iba1+ cell density (**a**) and percentage of coverage (**b**) are attenuated in HAB-EE compared to HAB in the posterior DG. Representative images of Iba1 in the anterior DG, scale bar 100 µm (**c**). Data are presented as mean ± SEM. *n* = 12–13. ** *p* < 0.01, *** *p* < 0.001.

**Figure 4 ijms-23-13052-f004:**
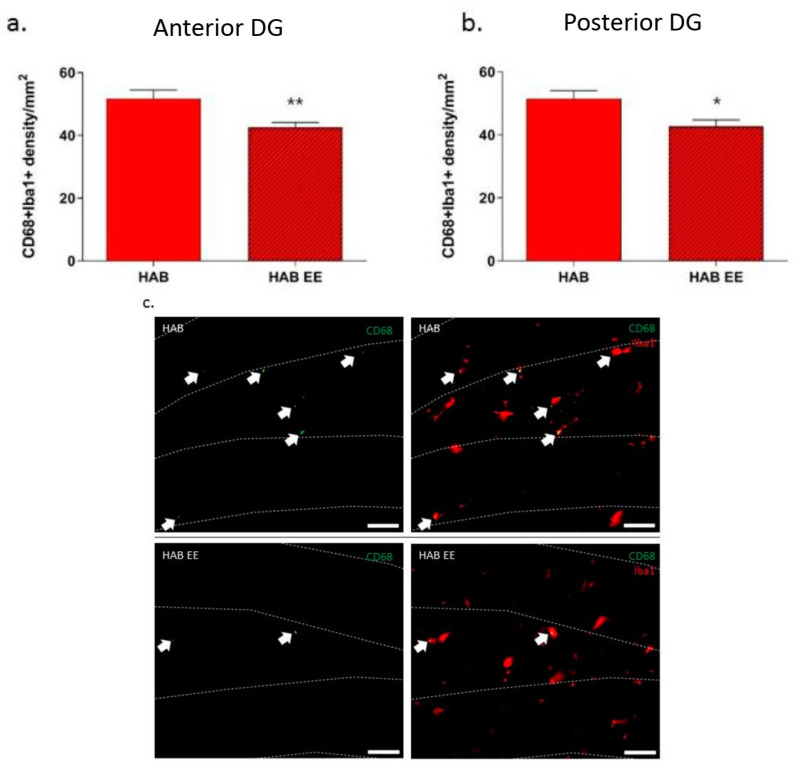
CD68 expression in Iba1+ cells in the anterior and posterior dentate gyrus (DG) of HAB is attenuated by EE housing. CD68 + Iba1+ cell density is significantly reduced in the anterior DG (**a**) and posterior DG (**b**) of HAB-EE compared to HAB. Representative images of Iba1 and CD68 in the anterior DG, outline is DAPI; scale bar 50 µm (**c**). Data are presented as mean ± SEM. *n* = 12–13. * *p* < 0.05, ** *p* < 0.01.

**Figure 5 ijms-23-13052-f005:**
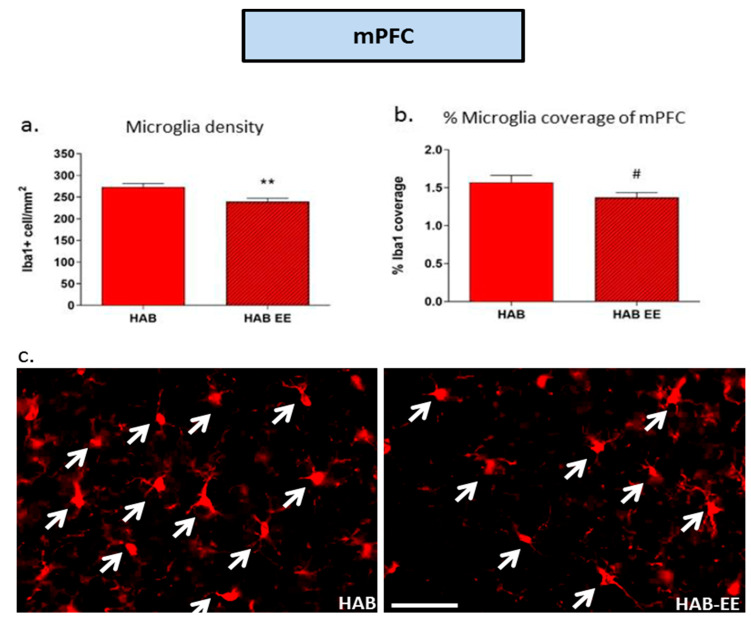
Iba1-related parameters in the medial prefrontal cortex (mPFC) were altered in HAB by EE housing. Iba1+ cell density (**a**) and percentage of coverage are attenuated in HAB-EE compared to HAB in the mPFC (**b**). Representative images of Iba1 in the mPFC indicated by arrows, scale bar 50 µm (**c**). Data are presented as mean ± SEM. *n* = 12–13. # *p* < 0.05, ** *p* < 0.01.

**Figure 6 ijms-23-13052-f006:**
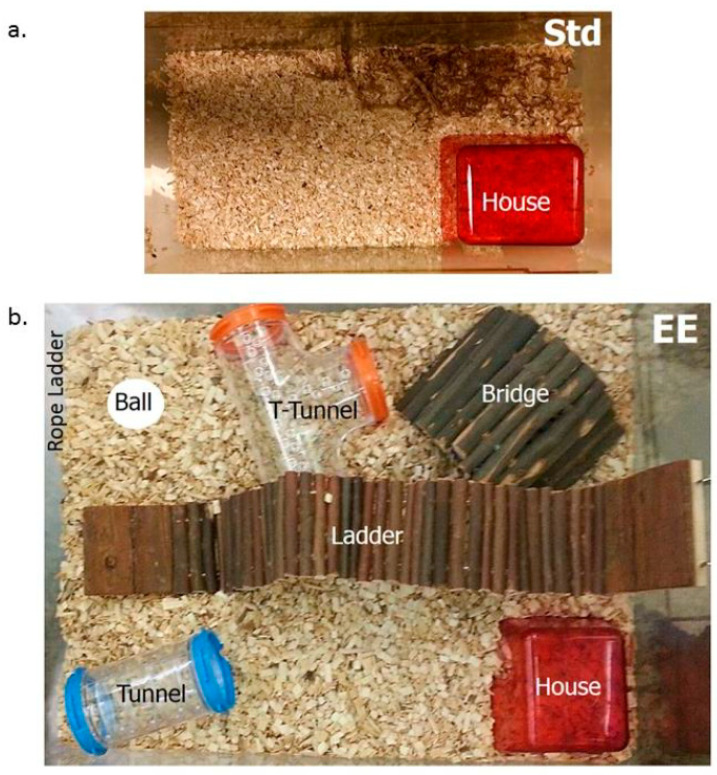
A representative image of the housing condition in a standard environment (**a**) or an enriched environment (EE) (**b**).

## Data Availability

Data are contained within the article.

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
