# Peer review of "Enriched Environment Attenuates Enhanced Trait Anxiety in Association with Normalization of Aberrant Neuro-Inflammatory Events"

_ijms, 2022, doi:10.3390/ijms232113052_

Round 1
Reviewer 1 Report
In the manuscript entitled „Enriched Environment Attenuates Enhanced Trait-Anxiety in Association with Normalization of Aberrant Neuro-Inflammatory Events” by Anupam Sah et al., the authors combined a mouse model for high trait anxiety (HAB mice) with environmental enrichment (EE) to assess whether the latter is able to ameliorate increased microglia activation, which is typical for this mouse line and, according to a recent study by the same group, causally involved in the high-anxiety phenotype of HABs. In detail, the authors show that EE, even when presented as late as during adulthood, can attenuate enhanced trait anxiety. The latter was associated with attenuation of enhanced microglial density in the dental gyrus and the medial-prefrontal cortex as well as microglial phagocytic activity within the DG. In extension of their previous study, these data suggest that besides pharmacological downregulation of microglia activation also positive behavioral therapies might be able to ameliorate overshooting microglia activation and, thus, represent a promising novel alternative or complimentary anxiolytic therapeutic approaches in specific subgroups of individuals predisposed to trait-anxiety.
Overall, the manuscript is excellently-written and adds important information to the field of PNI research. Therefore, this reviewer would support publication of this study in the International Journal of Molecular Sciences after the following concerns have been addressed:
-The authors should be a bit more careful with the interpretation of their findings and with indicating causality. Here are a couple of examples:
1) Abstract, l.24/25: “phagocytic activity” should be changed into “attenuation of co-labelled Iba1+CD68+ microglia density in the anterior DG”, as phagocytic activity was not measured. Was the Iba1+CD68+ microglia density in the PFC assessed as well?
2) Discussion, l. 250/251: “Taken together our results indicate that” should be changed into “Taken together our results support the hypothesis that” as causality was not shown in the present study
3) Discussion, l. 253: “Thus together with our pharmacology data, beneficial environment and or drugs targeting the microglial system can alleviate trait-anxiety symptoms in individuals via anti-inflammatory mechanisms in the CNS.” should be changed into “Thus together with our pharmacology data reported earlier, the findings of the current study suggest that beneficial environment targeting the microglial system can alleviate trait-anxiety symptoms in individuals via anti-inflammatory mechanisms in the CNS.”
4) Discussion, l.269 ff: “Thus, the evidence presented here indicates that microglia-mediated effects on neurogenesis in the DG may play a (partial) role in the maintenance of hyperanxiety in adult HAB mice.” should be changed into ““Thus, the evidence presented here suggests that microglia-mediated effects on neurogenesis in the DG may play a (partial) role in the maintenance of hyperanxiety in adult HAB mice.”
-Intro: The authors describe evidence indicating that rural vs. urban cues result in activation of different brain regions. As overshooting inflammation is in the focus of this study, the authors should consider mentioning studies showing increased inflammatory stress-responses in urban vs. rural individuals (i.e. Böbel et al., 2018 PNAS). The same group reported that repeated administrations of M. vaccae increases the number of iba1 positive microglia in the PFC of stressed mice and decreased stress-induced anxiety, which might be important for your the discussion of findings in NAB mice
-Intro, l.95/96: “high anxiety HAB mice”: as HAB stands for high anxiety-related behavior, this is redundant…the same in l.199 and 235…this should be adapted throughout the manuscript
-Results, l.115: The authors report only a trend towards increased locomotion in HAB EE mice, but the corresponding figure indicates a significant difference between the groups
-Results, LD test: the authors report a tendency towards an increased locomotion in HAB EE mice in the open field test. Therefore, it would be helpful to also report total distance travelled in the dark compartment of the LD box, as this parameter also is indicative of general locomotion.
-Discussion: first and second sentence seem to have been combined
-Discussion, l.193: environmental enrichment should be abbreviated (EE) as this abbreviation was introduced earlier already
-Discussion, l.205: “aslightly” should be changed into “a slightly”
-Discussion, l. 253: “and or“ should be changed into “and/or”
-Material&Methods, l.415: “elevated plus maze EPM test” should be changed into “elevated plus maze (EPM)
-Material&Methods, l.424: to what is Figure 4.1a referring to? A picture of an enriched cage would helpful for the reader
-Material&Methods, l.422ff: This reviewer was not able to figure out at which age of the experimental mice EE housing started. This would be important to mention, as you even emphasize the fact in the abstract, that the experimental mice were adult when exposed to EE housing.
Reviewer 2 Report
Enriched Environment Attenuates Enhanced Trait-Anxiety in Association with Normalization of Aberrant Neuro-Inflammatory Events
Anupam Sah, Sinead Rooney, Maria Kharitonova, Simone Sartori, Susanne A Wolf and Nicolas Singewald
Neuroinflammation has important roles in psychiatric disorders, including anxiety, with increase in microglial density and phagocytic activity in key regions of anxiety circuits. The authors in investigated whether environmental enrichment (EE) modulates neuroinflammation in high-anxiety model. The study shows that environmental enrichment attenuates trait anxiety thought attenuation of enhanced microglial density in the DG and medial-prefrontal cortex. The results also show that environmental enrichment reduced microglial phagocytic activity within the DG. The authors concluded that behavioral therapies could be promising as novel alternative or complimentary anxiolytic therapeutic approaches in specific subgroups of individuals predisposed to trait anxiety.
Although the authors' consent and observations are interesting, there are some points that require further justification and the manuscript is not suitable for publication without major revision.
Major comments:
- A more detailed protocol should be added describing the method used to count microglia, indicating how many slices/images you count, for example.
- The authors should perform at least one experiment with some control animals NABs, to demonstrated if EE enrichments reduces the anxiety to control levels, as well as the level of microglia activity.
- Another important point is that the authors did not measure any inflammatory cytokine, and this will be a proof of concept of reduction the neuroinflammation.
- Can the authors discuss the effect of EE in astrocytes, in this particular experimental protocol?
- In figure 2b, the images of Iba-1 immunohistochemistry are not representative of the graphs. Also, the images are at very lower resolution.
- In figure 3 and 5 the authors should add representative immunohistochemistry images.
- Can the authors observe in the images the shape of microglia (amoeboid or ramified), and discuss that?
